# Preparation and Characterization of Nanostructured Hollow MgO Spheres

**DOI:** 10.3390/ma12030537

**Published:** 2019-02-11

**Authors:** Jishuo Han, Guohua Li, Lin Yuan

**Affiliations:** 1School of High Temperature Materials and Magnesia Resources Engineering, University of Science and Technology Liaoning, Anshan 114051, China; hanjishuo0311@163.com; 2Ruitai Materials Technology Co., Ltd., China Building Materials Academy, Beijing 100024, China; yl@bjruitai.com

**Keywords:** template method, magnesite, carbon microspheres, nanostructured hollow MgO

## Abstract

Nanostructured hollow MgO microspheres were prepared by the template method. First, D-Anhydrous glucose was polymerized by the hydrothermal method to form a template. Second, a colorless solution was obtained by mixing magnesite with hydrochloric acid in a 1:2 proportion and heating in an 80 °C water bath for 2 h. Finally, the template from the first step was placed in the colorless solution, and the resulting precipitate was calcined at 550 °C for 2 h. The phase composition and microstructure of the calcined samples were characterized by means of X-ray diffraction (XRD), scanning electron microscopy (SEM) and transmission electron microscopy (TEM). The XRD results indicated that the main crystal is periclase. The SEM results indicates that the template carbon microsphere surface is smooth, and the its size is uniform and concentrated in the range of 100–200 nm. The diameters of the samples range from 60 to 90 nm, which is smaller than the size of the carbon microsphere. The TEM results indicates that the sample is hollow with a shell thickness of about 6–10 nm. The specific surface area of the calcined hollow sphere is 59.5 m²·g^−1^.

## 1. Introduction

Hollow microspheres are porous materials with special structures [1,2,3,4]. In recent years, hollow microspheres have been one of the frontier fields of material science. They have large specific surface areas and regularly adjustable pore structures, and are important catalytic, adsorption separation, and ion-exchange materials, which have been widely used in traditional petroleum refining, petrochemicals, fine chemicals, and other fields. In addition, these materials have also shown promising applications in the fields of optics, electricity, biology, medicine, sensing and nano-engineering [5]. Hollow microspheres currently have a variety of preparation methods, including the nozzle reactor [6], template [7], and sol-gel [8] methods. However, the preparation of hollow microspheres by the template method is simple, with a high repetition rate and controllable morphology. Three-dimensional multiparous (3DOM) LiMnPO_4_ as a cathode material for a lithium ion battery was synthesized by Hua Li et al. [9] with polymethyl methacrylate (PMMA) microspheres as templates prepared by a monodisperse polymerization method. Kong Qinglu et al. [10] reported a new approach, referred to as a “soft-to-hard templating” strategy, via the copolymerization of carbon source (dopamine) and silica source (tetraethyl-orthosilicate) for the synthesis of well dispersed N-doped mesoporous carbon nanospheres (MCNs). There are a large number of magnesia resources in southern Liaoning, which are widely used in metallurgical materials, refractories, chemical products, etc. Identifying means to utilize magnesia resources efficiently is becoming a popular research topic in the industry [11,12]. If nanostructured hollow microspheres can be prepared by natural minerals, the scope of application of magnesia will be widened. Due to its high activity surface and nano-effect, nano-magnesia can show new application functions, such as heat insulation, adsorption and molecular sieving [13]. Zhang Chao et al. [14] used Pickering emulsion droplets as templates to prepare nanostructured MgO hollow spheres and adsorbed Ni^2+^, but the specific surface area was only 28 m²·g^−1^. Yan Z et al. [15] reported on the formation of hollow MgO particles by excimer laser ablation of bulk Mg in water and aqueous solutions of sodium dodecyl sulfate (SDS) and sodium citrate (SC), but the operation was complicated and costly. Lisuzzo, L. et al. [16] focused on the structural and morphological features, performance, and ratios of inorganic nanotubes, considering the main strategies to prepare homogeneous colloidal suspensions in various solvent media as a special focus and crucial point for their uses as nanomaterials. Cavallaro, G. et al. [17] developed a novel green protocol for the deacidifying consolidation of waterlogged archaeological wood through aqueous dispersions of polyethylene glycol (PEG) 1500 and halloysite nanotubes containing calcium hydroxide.

In this paper, carbon-based adsorbent (carbon microspheres) prepared by the hydrothermal method of glucose was used as the template. The nanostructured hollow MgO microspheres were prepared by coating Mg^2+^ formed after the flotation of magnesia concentrate (Haicheng, Liaoning Province) and hydrochloric acid. The resulting products are used in the desulfurization or adsorption processes. The phase composition and microstructure of the samples were systematically characterized by X-ray diffraction (XRD), scanning electron microscopy (SEM), transmission electron microscopy and Brunner-Emmet-Teller (BET).

## 2. Materials and Methods

### 2.1. Materials

The D-anhydrous glucose (C_6_H_12_O_6_, AR) and absolute ethanol (C_2_H_6_O, AR, ≥99.7%) used in this experiment were of analytical grade and were purchased from Sinopharm Chemical Reagent Co., Ltd., Shanghai, China. Hydrochloric acid (36.0–38.0%) from Sinopharm Chemical Reagent Co., Ltd. and magnesite concentrate powder from Houying Co., Ltd., Haicheng, China, were used to prepare Mg^2+^. The particle size of magnesite concentrate was less than 0.074 mm. Table 1 shows the chemical composition of the magnesite concentrate.

### 2.2. Methods

45 g D-anhydrous glucose was dissolved in 250 mL deionized water and magnetically stirred at 450 r/min until the solution became clear. Then the solution was poured into a 300 mL reaction vessel and the reactor was incubated at 170 °C for 8 h. After the reaction, the dark brown liquid in the reaction kettle was poured out, washed with deionized water and absolute ethanol 3 times in succession, and suction filtered. The remaining material was dried at 90 °C for 10 h.

Magnesite concentrate (≤0.074 mm) was mixed with hydrochloric acid in 1:2 proportion in a 500 mL beaker and heated in an 80 °C water bath. After the reaction, the dark brown liquid in the reaction kettle was poured out and washed until no bubbles were formed. The precipitate was filtered out to produce a colorless solution. A quantity of 2 g of the prepared template carbon ball was added to the colorless solution obtained in the previous step, sonicated for 1 h, and then magnetically stirred for 3 h. After washing with deionized water 3 times, the dark-brown precipitate was obtained. After drying at 80 °C for 6 h, the sample was calcined at 550 °C for 2 h in an air atmosphere.

### 2.3. Theoretical Design

The template carbon microspheres were polymerized by the hydrothermal method using D-anhydrous glucose (C_6_H_12_O_6_, AR) as precursors [18]. Anhydrous glucose decomposes under hydrothermal conditions to form furfural aldehydes, oligosaccharide, and soluble monomers of some small molecules. These substances are condensed by intermolecular dehydration, and then through intramolecular dehydration to form C=O and C=C. Finally, carbon microspheres with hydrophilic shells (hydroxyl, carbonyl, carboxyl, ester) were formed, which can attract ions in solution, such as metal ions. The diameter of these spheres can be regulated by adjusting the hydrothermal time and the concentration of the glucose solution, and the whole process is green and chemical-free [19]. The nanostructured hollow MgO microspheres were prepared by the template method. Carbon microspheres formed by polymerization of D-anhydrous glucose as precursors were dispersed in the Mg^2+^ solution, which was prepared by reaction of magnesia concentrate with hydrochloric acid. Mg^2+^ was adsorbed on the surface of the carbon sphere by –OH and C=O, and deposited on the surface of the carbon sphere, then the carbon sphere template was removed to form a hollow spherical structure by calcination. A schematic diagram of the experimental synthesis of the MgO hollow microspheres is shown in Figure 1.

### 2.4. Determination of Decarbonization Temperature

Thermal decomposition of these carbon-adsorbed Mg^2+^ microspheres was studied by thermogravimetry-differential scanning calorimetry (TG-DSC, NETZSCH-Gerätebau GmbH, Shanghai, China). Figure 2 shows the thermograms obtained by heating the samples from room temperature to 1400 °C at 10 °C/min in air. As may be seen, between 250 °C and 400 °C, there are two exothermic peaks at which oxidation of the template carbon microspheres takes place, with a total mass loss of 89.22%. There are two exothermic peaks at 250 °C to 400 °C, while the mass loss of the sample is 85.22%. The first exothermic peak appears at 304.2 °C and the second exothermic peak appears at 397.3 °C, but the slope of the two exothermic peaks is different, which was attributed to the dehydroxy dehydration at the same time as the oxidation of carbon microspheres in the first part. With the appearance of some endotherms, the oxidation heat release of some carbon microspheres is offset, so the first exothermic peak is not sharper than the second exothermic peak. Dehydration occurs at the exothermic peak of the first part, and carbon oxidation occurs at the exothermic peak of the second part. With the increase of temperature, dehydration is completely removed, and the carbon microspheres continue to be oxidized. Between 400 °C and 500 °C, the absorbed magnesium ions oxidize to form magnesium oxide. When the temperature exceeds 550 °C, the mass does not change. Therefore, the temperature at which the carbon ball template is removed is determined to be 550 °C.

### 2.5. Characterization

Phases in the products were identified by X-ray diffraction (XRD, PANalytical X’Pert Powder diffractometer, PANalytical B.V., Almelo, The Netherlands) with Cu Kα radiation (λ = 1.5406 Å). Field emission scanning electron microscopy (SEM, ZEISS ΣIGMA, Carl Zeiss AG, Jena, Germany) and transmission electron microscopy (TEM) operated at 200kV (JEOL JEM2100, JEOL, Tokyo, Japan) were used to examine the phase composition and microstructure of the products. The specific surface area and gas adsorption isotherms of the adsorbent were tested using a surface area and pore size analyzer (BET, JW-BK112, JWGB SCI&TECH. Co., Ltd., Shanghai, China,) using nitrogen adsorption with a degassing temperature of 200 °C. Thermal analysis was tested using the synchronous thermal analyzer (Thermogravimetric Analysis-Differential Scanning Calorimetry (TG-DSC), STA 449 F3 Jupiter®, NETZSCH-Gerätebau GmbH, Shanghai, China).

## 3. Results

### 3.1. Structures of Carbon-Based Adsorbent Microspheres

Figure 3 shows the SEM image of the carbon microspheres. It shows that the surface of the carbon microspheres is smooth, the size is uniform, and the particle size is concentrated in the range of 100–200 nm.

### 3.2. Structures of the MgO Hollow Microspheres

Figure 4 shows the XRD pattern of the product after calcination. Compared with the standard MgO XRD chart, in which the two peaks are at the same location, the main peaks in Figure 4 are at 42.916° and 62.302°, so the calcined crystalline phase is periclase and the crystal development is incomplete. There are many defects.

The obtained hollow MgO microspheres were characterized by SEM, and the results are shown in Figure 5a,b. It can be seen in Figure 5a that the surface of each microsphere is smooth. The size of the hollow microspheres is 60–90 nm, which is greatly reduced compared to the diameter of the carbon spheres. The reason for this phenomenon is that the Mg^2+^ deposited onto the surface of the carbon sphere in the solution are deposited onto the surface of the carbon sphere template according to the heterogeneous nucleation theory and, then, the shrinkage and aggregation of the microcrystalline particles in the heat treatment process forms a branched net structure, resulting in a reduction in particle size. The existence of broken microspheres—that is, experimentally designed hollow structures—can be seen in Figure 5b. In our case, however, the hollow MgO microspheres synthetized by magnesia are little different from those synthetized by analysis grade MgCl_2_·6H_2_O in previous work in morphology [20], and we have the smaller aperture, reaching the nano-level.

The obtained hollow MgO microspheres were further characterized by TEM, as shown in Figure 6a,b. In Figure 6a, it can be seen that the sample is a hollow structure with a spherical shell thickness of about 6–10 nm. The hollow structure was formed in the sphere, and the structure was broken in the MgO microspheres. The hole radius of the hollow part was about 25 nm, which was consistent with the SEM results. The volume of the hole can be calculated by the spherical volume formula as 0.065 mm^3^. From the high-magnification TEM image shown in Figure 6b, the microspheres were composed of MgO nanoparticles and the porous structure was clearly observed. There was also a gap between the microcrystals, and the pore size was about 5–6 nm.

Nitrogen adsorption–desorption isotherms and the pore size distribution of the hollow MgO microspheres are shown in Figure 7. The isotherm shown in Figure 7a is type IV with H1 hysteresis loops [21]. In the low-pressure zone, the adsorption line coincides with the desorption line, and the adsorption in the low relative-pressure zone indicates that the material has a certain micropore structure. However, there is a hysteresis loop in the adsorption and desorption of the medium- and high-pressure zones, characteristic of mesoporous materials. From Figure 7b, the pore size distribution with centers of about 6 nm and 25 nm was observed, which were consistent with the TEM results. The specific surface area was 59.5 m²·g^−1^. In contrast, the specific surface area of the hollow magnesia ball developed by Zhang Chao et al. [22] was 28 m²·g^−1^.

## 4. Conclusions

D-anhydrous glucose as a precursor polymerization of carbon microspheres was formed as a template, after the flotation of magnesite powder and hydrochloric acid formed by the reaction of Mg^2+^ as the shell, and nanostructured hollow MgO microspheres were prepared. We can draw the following conclusions:(1)The calcination temperature of the samples is determined to be 550 °C by thermogravimetry-differential scanning calorimetry.(2)The surface of the carbon microspheres is smooth; the size is uniform and the particle size distribution is concentrated in the range of 100–200 nm.(3)The MgO microspheres have a hollow structure and periclase composition. The diameter of the sphere is concentrated in the range of 60–90 nm; the diameter of the hollow cavity is 50–60 nm; and the thickness of the spherical shell is 6–10 nm.(4)The surface of the spherical shell is composed of many crystallites and the microcrystalline gap in the surface of the spherical shell is 5–6 nm.(5)The specific surface area is about 59.5 m²·g^−1^.

## Figures and Tables

**Figure 1 materials-12-00537-f001:**
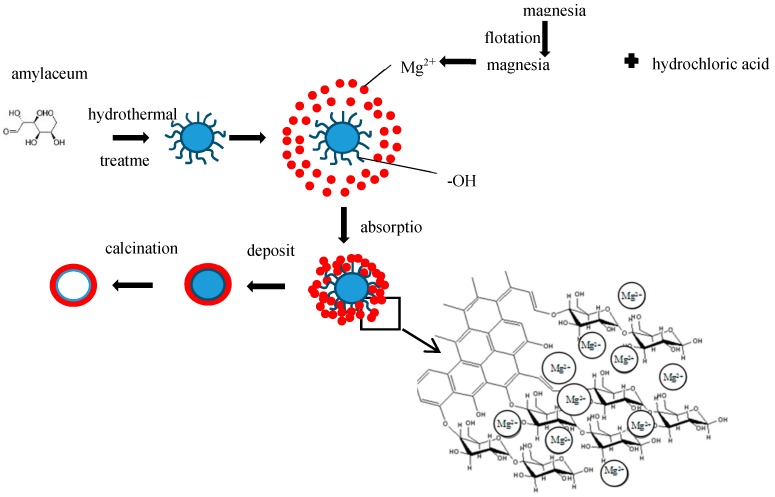
Schematic illustration of the preparation of hollow MgO microspheres.

**Figure 2 materials-12-00537-f002:**
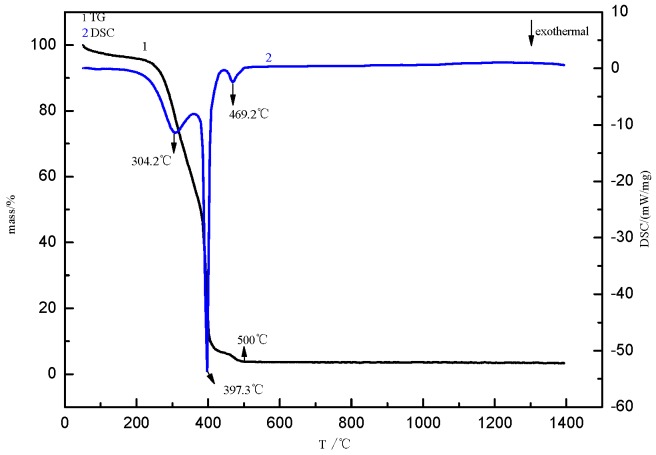
Thermogravimetry-differential scanning calorimetry (TG-DSC) curve of Mg^2+^ loaded with carbon microspheres.

**Figure 3 materials-12-00537-f003:**
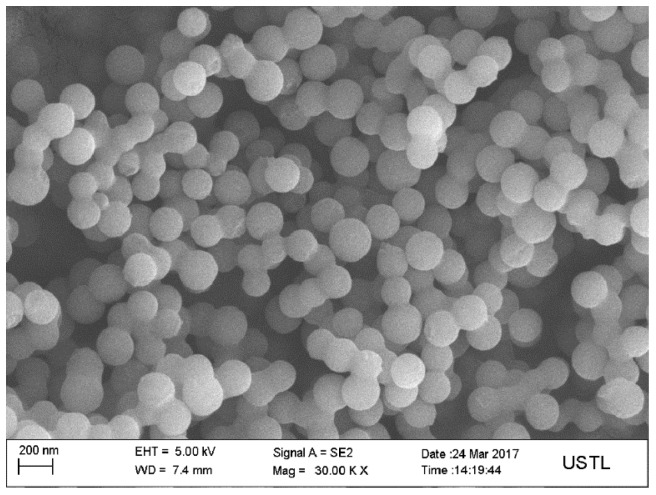
Scanning electron microscopy (SEM) images of the carbon microspheres (×30,000).

**Figure 4 materials-12-00537-f004:**
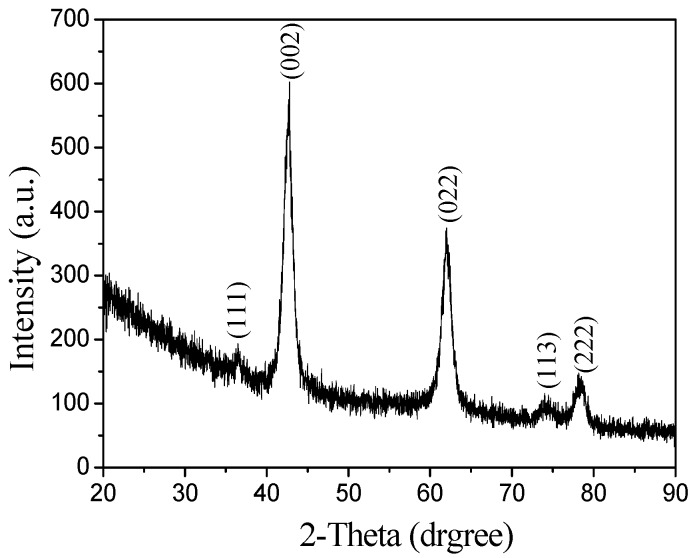
X-ray diffraction (XRD) pattern of the sample after calcination.

**Figure 5 materials-12-00537-f005:**
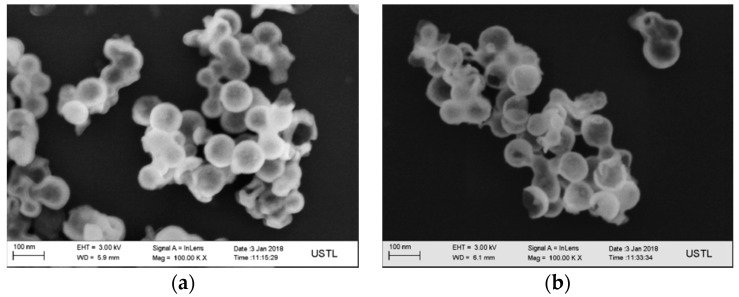
SEM images of the sample calcined in an air atmosphere at 550 °C (×100,000) (**a**): SEM images of intact magnesia hollow sphere (**b**): SEM images of cracked magnesia hollow sphere.

**Figure 6 materials-12-00537-f006:**
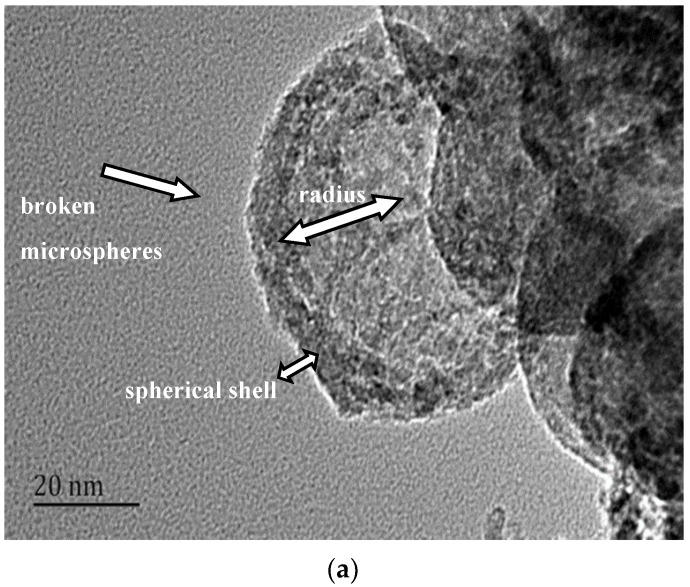
TEM images of the sample. (**a**): TEM images of cracked magnesia hollow sphere; (**b**): TEM images of the surface of cracked magnesia hollow sphere.

**Figure 7 materials-12-00537-f007:**
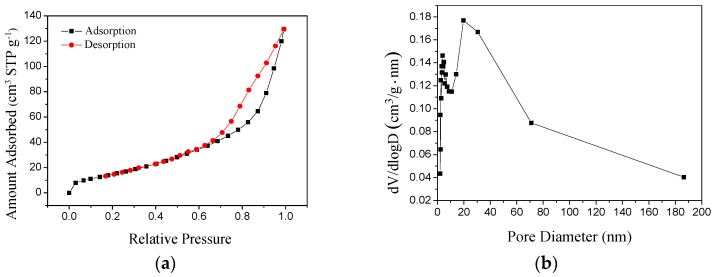
Nitrogen absorption–desorption isotherm (**a**) and the pore size distribution (**b**) of the sample.

**Table 1 materials-12-00537-t001:** Chemical composition of the concentrate magnesite.

Ingredient	MgO	CaO	SiO_2_	Fe_2_O_3_	Al_2_O_3_	Calcination Reduction (IL)
ω	47.69	0.49	0.18	0.35	0.07	51.22

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
