# Peer review of "Preparation and Characterization of Nanostructured Hollow MgO Spheres"

_materials, 2019, doi:10.3390/ma12030537_

Round 1
Reviewer 1 Report
The paper deals with the morphological and thermal characterization of nanostructured hollow MgO microspheres. The topic falls within the topic of Materials and the data are properly presented and dicussed. I recommend the publication after the following revisions:
- Paragraph 2.4. As presented by the authors, it looks that the exothermic peaks related to the oxidation process were determined by thermogravimetry. Actually, TG analysis did not provide information on the thermodynamics. These exothermic peaks were determined by Differential Scanninfg Calorimetry. According to this consideration, I recommend to revise the discussion in paragraph 2.4.
- The working distance for SEM/TEM analyses should be added in the Experimental section.
- Figure 2. The labels for y-axes should be “mass / %” and “heat flow / W/g”
- In general,the quality of SEM/TEM images is poor. Please improve it.
- Introduction could be updated by quoting other hollow inorganic and organic particles with different shape, such as clay nanotubes [ACS Appl Mater Interfaces. 10, 2018, 27355-27364; Appl. Sci. 2018, 8(7), 1068], organo-silica [Molecules 2019, 24(2), 332] and wool-like nanoparticles [Pharmaceutics 2018, 10(2), 52] , as templates to create functional nanostructures.
- Is it possibile to conduct a statistical analysis of the nanoparticles sizes from SEM/TEM micrographs?
Author Response
Response to Reviewer 1 Comments
Thank you very much foy your very valuable advices on my manuscriptduring you busy schedule. I have carefully thought about and changed every suggestions. And revised the introduction, conclusion, and summarysections, and polished the article language, i hope that you can consider it. Thang you again for your review.
Point 1: Paragraph 2.4. As presented by the authors, it looks that the exothermic peaks related to the oxidation process were determined by thermogravimetry. Actually, TG analysis did not provide information on the thermodynamics. These exothermic peaks were determined by Differential Scanninfg Calorimetry. According to this consideration, I recommend to revise the discussion in paragraph 2.4.
Response 1: I have modified paragraph 2.4 as you suggest, and re-expain the results in detail, the results of the amendments can be found in the manuscript. Please check.
Point 2: The working distance for SEM/TEM analyses should be added in the Experimental section.
Response 2: Due to different measurement requirements, the working distance is variable. I have marked in each SEM/TEM pictures title. Please check.
Point 3: Figure 2. The labels for y-axes should be “mass / %” and “heat flow / W/g”
Response 3: I have corrected the labels for y-axes. Please check.
Point 4: In general,the quality of SEM/TEM images is poor. Please improve it.
Response 4: I hve improved the pictures clarity. Please check.
Point 5: Introduction could be updated by quoting other hollow inorganic and organic particles with different shape, such as clay nanotubes [ACS Appl Mater Interfaces. 10, 2018, 27355-27364; Appl. Sci. 2018, 8(7), 1068], organo-silica [Molecules 2019, 24(2), 332] and wool-like nanoparticles [Pharmaceutics 2018, 10(2), 52] , as templates to create functional nanostructures.
Response 5: I have read the relevant articles and credited according to your suggestion, and revised the introduction. Please check.
Point 6: Is it possibile to conduct a statistical analysis of the nanoparticles sizes from SEM/TEM micrographs?
Response 6: Use SEM measurement size function to measure the aperture size is allowed. Please check.

Reviewer 2 Report
The paper presents preparation and characterization of nanostructured hollow MgO microspheres by magnesia. These issues are interesting, but some remarks concerning the preparation of the manuscript should be taken into consideration while revising the paper:
- All references should be adjusted to the Materials rules. This applies to the references in the References section. Please also pay attention to: dots, commas, upper indices, spaces.
- The paper presents test results for only one material. Please compare the received material with other similar materials based on literature data or previous own research.
- The paper describes a high specific surface area (59.5 m2 / g) of the obtained material. This is a relative concept. Compare this material with other materials which have similar properties.
- Analyzing the isotherm shown in the work, it is known that the material tested is mesoporous, with only a small amount of micropores. I am asking for the determination and presentation of the total pore volume, mesopore volume and the volume of micropores in the work.
- Fig. 2 and a discussion of this figure should appear in Results.
- What is the purpose of the research being carried out? What could be the use of these materials? Please, fill in this information at work.
Author Response
Response to Reviewer 2 Comments
Thank you very much foy your very valuable advices on my manuscriptduring you busy schedule. I have carefully thought about and changed every suggestions. And revised the introduction, conclusion, and summarysections, and polished the article language, i hope that you can consider it. Thang you again for your review.
Point 1: All references should be adjusted to the Materials rules. This applies to the references in the References section. Please also pay attention to: dots, commas, upper indices, spaces.
Response 1: I have made careful formatting changes to the citation as required, please check.
Point 2:The paper presents test results for only one material. Please compare the received material with other similar materials based on literature data or previous own research.
Response 2: In the introduction we have made some comparisons and compared similar materials in the experimental results, please check.
Point 3: The paper describes a high specific surface area (59.5 m2 / g) of the obtained material. This is a relative concept. Compare this material with other materials which have similar properties.
Response 3: As you suggested, i have compared other materilas, see 184 lines, please check.
Point 4: Analyzing the isotherm shown in the work, it is known that the material tested is mesoporous, with only a small amount of micropores. I am asking for the determination and presentation of the total pore volume, mesopore volume and the volume of micropores in the work.
Response 4: I have calculated the volume of the hole based on your contact, see line 168, please check.
Point 5: Fig. 2 and a discussion of this figure should appear in Results.
Response 5: According to your suggestion, the discussion in Fig. 2 already appears in the conclusion, please check.
Point 6: What is the purpose of the research being carried out? What could be the use of these materials? Please, fill in this information at work.
Response 6: In hte introduction, we have introduced the working environment of such aterials and the significance of studying such materials, and explained that these materials mainly play the role of adsorption.

Reviewer 3 Report
This paper describes the synthesis route of MgO hollow spheres having nanostructured walls. In my opinion, the method described in the paper is interesting, and can be useful to other researchers. However, the paper should be revised before publication. Please see some comments below.
From what I learned, it appears that the synthesized spheres can be called nano-sized spheres and not microspheres. I suggest changing the paper title to "Preparation and characterization of nanostructured hollow MgO spheres". "By magnesia" sounds a bit out of place in the title. The Abstract of the paper and Conclusions should be carefully revised in terms of terminology. What do you mean by "the size is symmetrical"? "Symmetrical" is used for shapes, not sizes.
Please explain why you selected D-anhydrous glucose as a precursor of carbon spheres.
"Mg2+ oxidizes to form MgO" is not correct, as magnesium in MgO has a charge of 2+. So, there is no oxidation. Do you observe formation and decomposition of Mg(OH)2?
Please provide XRD of the material before calcination.
The sentence "However, the process of the oxidation of carbon microspheres is a slow process, so the quality changes in this stage are fine" should be revised to clarify its meaning. What is meant by "fine"?
Caption to Fig. 4 should be revised. Please provide details of the sample preparation procedure, otherwise it is not clear what sample has been analyzed.
"Figure 5. SEM images of the sample fires in an air atmosphere at 550℃". Should be "calcined", not "fires".
What is meant by "we have the smaller aperture" in the context of MgO spheres?
Author Response
Response to Reviewer 3 Comments
Thank you very much foy your very valuable advices on my manuscriptduring you busy schedule. I have carefully thought about and changed every suggestions. And revised the introduction, conclusion, and summary sections, and polished the article language, i hope that you can consider it. Thang you again for your review.
Point 1: From what I learned, it appears that the synthesized spheres can be called nano-sized spheres and not microspheres. I suggest changing the paper title to "Preparation and characterization of nanostructured hollow MgO spheres". "By magnesia" sounds a bit out of place in the title. The Abstract of the paper and Conclusions should be carefully revised in terms of terminology. What do you mean by "the size is symmetrical"? "Symmetrical" is used for shapes, not sizes.
Response 1: I have modified the title as per your suggestion and changed the word to describe the aperture size. Please check.
Point 2:Please explain why you selected D-anhydrous glucose as a precursor of carbon spheres.
Response 2: Since D-anhydrous glucose itself is green and healthy and does not require any chemical solvents to be added, and the resulting pores are more rounded and smaller in diameter.
Point 3: "Mg2+ oxidizes to form MgO" is not correct, as magnesium in MgO has a charge of 2+. So, there is no oxidation. Do you observe formation and decomposition of Mg(OH)2?
Response 3: I have made changes to the suggestions you have given. Please check.
Point 4: Please provide XRD of the material before calcination.
Response 4: Sorry, we are unable to provide XRD of the material before calcination, because we did not do XRD before calcinating during the experiment. Appologize again.
Point 5: The sentence "However, the process of the oxidation of carbon microspheres is a slow process, so the quality changes in this stage are fine" should be revised to clarify its meaning. What is meant by "fine"?
Response 5: According to your suggestion, we have modified the original text and explained here why the changes have not changed much.
Point 6: Caption to Fig. 4 should be revised. Please provide details of the sample preparation procedure, otherwise it is not clear what sample has been analyzed.
Response 6: I have modified the title 4 and emphasized that it is a figure of the sample after calcinating. The preparation of the sample has been given in 2.2 Methods. Please check.
Point 7:"Figure 5. SEM images of the sample fires in an air atmosphere at 550℃". Should be "calcined", not "fires".
Response 7: I have made changes. Please check.
Point 8:What is meant by "we have the smaller aperture" in the context of MgO spheres?
Response 8: We have a relatively smaller aperture compared to those synthetized by analysis grade MgCl2·6H2O in previous work in morphology. Please check.

Round 2
Reviewer 1 Report
The revised MS is suitable for publication in the present form.
Reviewer 3 Report
I think that the paper has been greatly improved relative to its previous version.
The required clarifications have been added. The language of the paper has been much improved.